# LEARNING DISTRIBUTIONS GENERATED BY SINGLE-LAYER ReLU NETWORKS IN THE PRESENCE OF ARBITRARY OUTLIERS

## ABSTRACT

We consider a set of data samples such that a constant fraction of the samples are arbitrary outliers and the rest are the output samples of a single-layer neural network (NN) with rectified linear unit (ReLU) activation. The goal of this paper is to estimate the parameters (weight matrix and bias vector) of the NN assuming the bias vector to be non-negative. Our proposed method is a two-step algorithm. We first estimate the norms of the rows of the weight matrix and the bias vector using the gradient descent algorithm. Here, we also incorporate either the median or the trimmed mean based filters to mitigate the effect of the arbitrary outliers. Next, we estimate the angles between any two row vectors of the weight matrix. Combining the estimates of the norms and the angles, we obtain the final estimate of the weight matrix. Our main contribution is the sample complexity for robust estimation rather than the algorithm itself. Here, we prove that $\Omega(\frac{1}{\epsilon p^4} \log \frac{d}{\delta})$ samples are sufficient for our algorithm to estimate the NN parameters within an error of $\epsilon$ with probability $1 - \delta$ when the probability of a sample being uncorrupted is $p$ and the problem dimension is $d$. Our theoretical and simulation results provide insights on how the estimation of the NN parameters depends on the probability of a sample being uncorrupted, the number of samples, and the problem dimension.

## 1 INTRODUCTION

A fundamental challenge in machine learning and statistics is to estimate a high dimensional distribution given a set of observed data samples from the distribution. One solution technique is the deep generative method that models the unknown distribution as the output distribution of a neural network (NN) when the input of the NN is drawn from a known distribution like standard Gaussian which is motivated from the case of image generation by generative adversarial networks (GANs) (Goodfellow et al. (2014); Arjovsky et al. (2017); Radford et al. (2015); Kingma & Welling (2013); Van Oord et al. (2016)). Then, the unknown distribution is estimated by learning the NN parameters from the data samples. Several approaches like GANs (Goodfellow et al. (2014); Arjovsky et al. (2017); Radford et al. (2015)), variational autoencoders (Kingma & Welling (2013)), and autoregressive models (Van Oord et al. (2016)) have been proposed to train the NN. However, they do not provide theoretical guarantees for the parameter learning. So, we address the open problem of computing the sample complexity for the parameter estimation of a single-layer NN with rectified linear unit (ReLU) activation using a corrupted dataset in the unsupervised learning framework. Here, the corrupted dataset refers to the model wherein the output samples consists of a fraction of arbitrary outliers introduced by an adversary (Byzantine). Motivated by the deep generative models, we consider the unsupervised learning framework, i.e., we assume the knowledge of the output samples but, the corresponding inputs are unknown and drawn from the standard Gaussian distribution.

We start with a brief review of the related literature. The NN parameter estimation has been considered under both supervised and unsupervised learning frameworks. The estimation rely on the stochastic gradient descent (SGD)-based algorithms (Goel et al. (2018); Allen-Zhu et al. (2019); Chen et al. (2020); Oymak (2019); Mazumdar & Rawat (2018); Wu et al. (2019); Lei et al. (2020)); or the gradient descent (GD)-based approach (Cao & Gu (2019); Du et al. (2019)). A major shortcoming of these works is that they assume that the available samples are not corrupted by noise or outliers. Several works in the literature have addressed the problem of corrupted samples in the

context of supervised learning. The existing works have considered the learning of the parameters of a NN with ReLU activation using both SGD-based (Bakshi et al. (2019); Goel et al. (2019); Mukherjee & Muthukumar (2020)) and GD-based algorithms(Zhang et al. (2019); Frei et al. (2020); Vempala & Wilmes (2019)). However, to the best of our knowledge, the estimation of a NN in the presence of noise or outliers in the unsupervised learning framework has not been studied in the literature.

Our setting of unsupervised parameter estimation is also related to the area of robust statistics as described below. We recall that the goal of our paper is to estimate the parameters of a NN using corrupted output samples assuming that the input distribution is Gaussian. Mathematically, this problem is equivalent to estimating the parameters of a truncated Gaussian distribution (see Section 2 for details). The area of robust statistics also deals with the estimation of high dimensional distributions like Gaussian, Gaussian product, and Gaussian mixture distributions where a constant fraction of the samples were corrupted by noise (Huber (2004); Hampel et al. (2011); Lai et al. (2016); Diakonikolas et al. (2019); Diakonikolas & Kane (2019); Kane (2021)). However, in this paper, we consider parameter estimation from truncated Gaussian samples in the adversarial setting which has not been studied in the literature.

We present our major contributions as the following.

- *Parameter Estimation (Sections 2 and 3):* We estimate the parameters (weight matrix and bias vector) of a single-layer NN with ReLU in the unsupervised learning framework where each output sample can potentially be an arbitrary outlier with a fixed probability. We propose an estimation algorithm that has two steps: (i) we estimate the row norms of the weight matrix and the bias vector using GD along with either the median or trimmed mean-based filters to mitigate the effect of arbitrary outliers, and (ii) we estimate the angles between any two row vectors of the weight matrix using a simple geometric result (Williamson & Shmoys, 2011, Lemma 6.7).

- *Theoretical guarantees (Section 4):* We show that the proposed algorithm requires $\Omega(\frac{1}{\epsilon p^4} \log \frac{d}{\delta})$ samples to estimate the network parameters within an error of $\epsilon$ with probability $1 - \delta$ when the probability of a sample being uncorrupted is $p$. Here, $\epsilon > 0$ represents the estimation error.

- *Empirical validation (Section 5):* We evaluate the performance of our algorithm empirically in terms of its relation to the probability of a sample being uncorrupted, number of samples, and dimension (see Figs. 1 and 2). We see that our proposed filtering schemes ensure robustness to the arbitrary outliers. Also, the performance of our algorithm improves with the increase in the probability of a sample being uncorrupted and the number of samples, as expected. Further, we note that the estimation error increases slowly as the dimension grows for a fixed number of samples in the presence of arbitrary outliers. These observations from the empirical results are consistent with our theoretical results.

To summarize, in this paper, we estimate the parameters of a single-layer NN with ReLU activation in the presence of arbitrary outliers. The results obtained in this paper provide insights to generalize to a multi-layer NN with various activation functions and obtain generalization bounds.

## 2 LEARNING A NEURAL NETWORK MODEL USING CORRUPTED OUTPUT SAMPLES

We consider a single-layer ReLU NN. Let the weight matrix of the NN be denoted by $\mathbf{W} \in \mathbb{R}^{d \times k}$ and the bias vector by $\mathbf{b} \in \mathbb{R}^d$. The input to the NN is denoted by the latent variable $\mathbf{z} \in \mathbb{R}^k$. We assume that the variable $\mathbf{z}$ is drawn from the standard Gaussian distribution. The Gaussianity assumption is for mathematical tractability, and it is motivated by a popular generative approach to estimate a high-dimensional distribution from observed samples in the case of image generation by GANs Goodfellow et al. (2014); Radford et al. (2015); Arjovsky et al. (2017). Thus, the output of the network is the random vector $\mathbf{x} \in \mathbb{R}^d$ given by

$$\mathbf{x} = \text{ReLU}(\mathbf{Wz} + \mathbf{b}), \text{ where } \mathbf{z} \sim \mathcal{N}(\mathbf{0}, \mathbf{I}_k), \tag{1}$$

where $\mathcal{N}(\mathbf{0}, \mathbf{I}_k)$ denotes the Gaussian distribution with mean $\mathbf{0}$ and covariance matrix $\mathbf{I}_k$ which is the identity matrix of size $k \times k$. Let the distribution of the random vector $\mathbf{x}$ be denoted by $\mathcal{D}(\mathbf{W}, \mathbf{b})$. Then, our goal is to estimate the unknown parameters $\mathbf{W}$ and $\mathbf{b}$ of the distribution $\mathcal{D}(\mathbf{W}, \mathbf{b})$ using

the knowledge of $n$ output samples $\mathbf{x}_1, \mathbf{x}_2, \ldots, \mathbf{x}_n$. Here, we assume that a fraction of the samples are arbitrary outliers. The distribution of the observed samples are modeled using the *Huber's p-contamination model* [1] defined as the following.

**Definition 2.1** (Huber's $p$-Contamination Model Huber (1964)). *The observed samples are said to be following Huber's contamination model if any given sample is drawn from the true distribution $\mathcal{D}$ and an arbitrary distribution $\mathcal{G}$ with probability $p$, and $1 - p$, respectively. In other words, the observed samples are drawn from the mixture distribution, $\mathcal{D}_p(\mathbf{W}, \mathbf{b})$ given by*

$$\mathcal{D}_p(\mathbf{W}, \mathbf{b}) = p\mathcal{D}(\mathbf{W}, \mathbf{b}) + (1-p)\mathcal{G}. \tag{2}$$

Note that if $\mathbf{b}$ has large negative values, then most of the output samples are zeros due to the ReLU operation. Further, if $\mathbf{W}$ is a zero matrix, then any negative coordinate of $\mathbf{b}$ cannot be identified as it is reset to zero after the ReLU operation. Further, in Wu et al. (2019), the authors have shown that if $\mathbf{b} \in \mathbb{R}^d$ then exponentially large number of samples are required to estimate the bias. This holds true in our case as in the presence of arbitrary outliers if $\mathbf{b}$ has large negative values, then we would require larger number of samples to estimate the bias vector $\mathbf{b}$ compared to that in Wu et al. (2019). Hence, we also assume $\mathbf{b}$ to be non-negative.

Our estimation problem is challenging due to two reasons: 1) ReLU operation is not invertible, and therefore, estimation of $\mathbf{W}$ and $\mathbf{b}$ utilizing maximum likelihood of the probability density function of the random vector $\mathbf{x}$ is intractable; 2) the distribution of the corrupted samples is unknown. We tackle these issues using a new formulation combining the GD algorithm and a filtering technique. Note that the GD algorithm is similar to that in Wu et al. (2019) wherein the authors use the SGD algorithm. The proposed algorithm is presented in the next section.

## 3 ESTIMATION OF PARAMETERS OF SINGLE-LAYER NN

To design our estimation algorithm, we first note that the weight matrix $\mathbf{W}$ may not be identifiable from the distribution $\mathcal{D}(\mathbf{W}, \mathbf{b})$. This is because of the fact that for any matrices $\mathbf{W}_1$ and $\mathbf{W}_2$ if $\mathbf{W}_1\mathbf{W}_1^\mathsf{T} = \mathbf{W}_2\mathbf{W}_2^\mathsf{T}$ then there exists a unitary matrix $\mathbf{Q}$ such that $\mathbf{W}_2 = \mathbf{W}_1\mathbf{Q}$. As $\mathbf{z} \sim \mathcal{N}(\mathbf{0}, \mathbf{I}_k)$, we have $\mathbf{Q}\mathbf{z} \sim \mathcal{N}(\mathbf{0}, \mathbf{I}_k)$. Hence, for any vector $\mathbf{b}$, we have $\mathcal{D}(\mathbf{W}_1, \mathbf{b}) = \mathcal{D}(\mathbf{W}_2, \mathbf{b})$ (Wu et al. (2019)). Since our goal is to learn the distribution, learning either $\mathbf{W}_1$ or $\mathbf{W}_2$ is sufficient. In short, we focus on the learnability of the underlying distribution and not the learnability of the NN parameters. Therefore, our proposed algorithm estimates $\mathbf{W}\mathbf{W}^\mathsf{T} \in \mathbb{R}^{d \times k}$ and $\mathbf{b} \in \mathbb{R}^d$ from the observed samples from $\mathcal{D}_p(\mathbf{W}, \mathbf{b})$.

Next, we note that $(i,j)$-th entry $(\mathbf{W}\mathbf{W}^\mathsf{T})(i,j)$ of the symmetric matrix $\mathbf{W}\mathbf{W}^\mathsf{T}$ is $\|\mathbf{W}(i,:)\|_2 \|\mathbf{W}(j,:)\|_2 \cos(\theta_{ij})$, where $\theta_{ij} = \theta_{ji}$ is the angle between vectors $\mathbf{W}(i,:)$ and $\mathbf{W}(j,:)$ which are the $i$-th and $j$-th rows of matrix $\mathbf{W}$, respectively. Also, note that the $\ell_p$ norm of a vector is given by $\|\mathbf{x}\|_p = (\sum_i |\mathbf{x}(i)|^p)^{1/p}$. Thus, we can construct the matrix $\mathbf{W}\mathbf{W}^\mathsf{T}$ using the row norms $\{\|\mathbf{W}(i,:)\|_2\}_{i=1}^d$ and angles $\{\theta_{ij}\}_{i,j=1}^d$.

Our estimation algorithm consists of two steps: 1) estimation of row norms of $\mathbf{W}$ and bias vector $\mathbf{b}$; and 2) estimation of angles between the row vectors of $\mathbf{W}$.

To estimate $\|\mathbf{W}(i,:)\|_2 \in \mathbb{R}$ and $\mathbf{b}(i) \in \mathbb{R}$, for $i \in [d] = \{1, 2, \ldots, d\}$, suppose $\mathbf{x} \sim \mathcal{D}(\mathbf{W}, \mathbf{b})$. Its $i$-th coordinate can be written as

$$\mathbf{x}(i) = \text{ReLU}(\mathbf{W}(i,:)^\mathsf{T}\mathbf{z} + \mathbf{b}(i)), \tag{3}$$

where $\mathbf{W}(i,:)^\mathsf{T}\mathbf{z} + \mathbf{b}(i) \sim \mathcal{N}(\mathbf{b}(i), \|\mathbf{W}(i,:)\|_2^2)$. Thus, to compute $\mathbf{b}(i)$ and $\|\mathbf{W}(i,:)\|_2^2$, it is enough to consider the $i$-th entries of the observed samples. Also, in (3), the ReLU operator sets $\mathbf{x}(i)$ to zero if $\mathbf{W}(i,:)^\mathsf{T}\mathbf{z} + \mathbf{b}(i)$ is negative. Consequently, we estimate $\mathbf{b}(i)$ and $\|\mathbf{W}(i,:)\|_2^2$ using the positive entries of the observed samples. Hence, estimation of $\mathbf{b}(i)$ and $\|\mathbf{W}(i,:)\|_2$ is equivalent to estimating the parameters of a one-dimensional normal distribution using positive samples, i.e., the samples that belong to the set $\mathbb{R}_{>0} := \{\mathbf{x}(i) \in \mathbb{R} : \mathbf{x}(i) > 0\}$. We use $\mu^* = \mathbf{b}(i)$ and $\sigma^{*2} = \|\mathbf{W}(i,:)\|_2$ to denote the parameters of this one-dimensional normal distribution. The estimation of $\mu^*$ and $\sigma^{*2}$ using corrupted samples is discussed next.

---

[1] Our algorithm and its analysis follow even if the outliers are not sampled from the same arbitrary distributions $\mathcal{G}$. However, we need to assume that all the true samples come from the same distribution $\mathcal{D}$.

**Algorithm 1:** Learning one-layer ReLU NN with outliers

**Input:** Samples $\mathbf{x}_1, \ldots, \mathbf{x}_n \in \mathbb{R}^d$
1 **for** $i \in [d]$ **do**
2     $\mathcal{X}_{>0} \leftarrow \{\mathbf{x}_m, m \in [n] : \mathbf{x}_m(i) > 0\}$
3     $\hat{\mathbf{v}} = \text{ProjGD}(\mathcal{X}_{>0})$
4     $\hat{\mathbf{\Sigma}}_{i,i} \leftarrow 1/\hat{\mathbf{v}}(1)$
5     $\hat{\mathbf{b}}(i) \leftarrow \max\{0, \hat{\mathbf{v}}(2)/\hat{\mathbf{v}}(1)\}$
6 **for** $i < j \in [d]$ **do**
7     Compute $\hat{\theta}_{ij}$ using (11)
8     $\hat{\mathbf{\Sigma}}(i,j) \leftarrow \sqrt{\hat{\mathbf{\Sigma}}(i,i)\hat{\mathbf{\Sigma}}(j,j)} \cos(\hat{\theta}_{ij})$
9     $\hat{\mathbf{\Sigma}}(j,i) \leftarrow \hat{\mathbf{\Sigma}}(i,j)$

**Output:** $\hat{\mathbf{\Sigma}} \in \mathbb{R}^{d \times d}, \hat{\mathbf{b}} \in \mathbb{R}^d$

**Algorithm 2:** ProjGD

**Input:** Samples $\mathcal{X}_{>0}$, Parameters $T, \gamma_t$, and $n_b$
1 Compute $\{\overline{\mathbf{x}_j(i)}\}_{j=1}^{n_b}$ from $\mathcal{X}_{>0}$ using (7)
2 $\tilde{\mathbf{g}}_\mathbf{x} \leftarrow \text{filter}\left(\{\overline{\mathbf{x}_j(i)}\}_{j=1}^{n_b}\right)$
3 $\mathbf{v}_0 \leftarrow \begin{bmatrix} 0 & 0 \end{bmatrix}$
4 **for** $t = 1, 2, \ldots, T$ **do**
5     $\mu \leftarrow \mathbf{v}_{t-1}(2)/\mathbf{v}_{t-1}(1)$
6     $\sigma^2 \leftarrow 1/\mathbf{v}_{t-1}(1)$
7     $\tilde{\mathbf{g}}_\mathbf{z} \leftarrow \begin{bmatrix} -(\breve{\sigma}^2 + \breve{\mu}^2)/2 & \breve{\mu} \end{bmatrix}^\mathsf{T}$ using (5) and (6)
8     $\tilde{\mathbf{g}}_t \leftarrow \tilde{\mathbf{g}}_\mathbf{x} + \tilde{\mathbf{g}}_\mathbf{z}$
9     $\tilde{\mathbf{v}}_t \leftarrow \mathbf{v}_{t-1} - \gamma_{t-1}\tilde{\mathbf{g}}_{t-1}$
10     $\mathbf{v}_t \leftarrow P(\tilde{\mathbf{v}}_t)$ using (10)

**Output:** $\mathbf{v}_T \in \mathbb{R}^2$

We determine the parameters of the univariate normal distribution using maximum likelihood estimation (Daskalakis et al. (2018)) given by $\arg\min_\mathbf{v} \bar{\ell}(\mathbf{v})$ where $\bar{\ell}(\mathbf{v})$ is the expected negative log-likelihood with respect to $\mathbf{x}(i)$ and $\mathbf{v} = [1/\sigma^2; \mu/\sigma^2] \in \mathbb{R}^2$. This optimization problem can be solved using a learning algorithm like GD or SGD. SGD utilizes only one sample for the gradient computation which introduces large variance due to the stochasticity and low reliability in the presence of arbitrary outliers. Thus, we use GD that utilizes all the samples for the gradient.

To derive our algorithm based on GD, we compute the gradient (Daskalakis et al. (2018)), $\nabla\bar{\ell}(\mathbf{v}) = \mathbb{E}_{\mathbf{x},\mathbf{z}}[\mathbf{g}(i)]$ where

$$\mathbf{g}(i) = \mathbf{g}_\mathbf{x}(i) + \mathbf{g}_\mathbf{z}(i) = \begin{bmatrix} \mathbf{x}^2(i)/2 & -\mathbf{x}(i) \end{bmatrix}^\mathsf{T} + \begin{bmatrix} -\mathbf{z}^2(i)/2 & \mathbf{z}(i) \end{bmatrix}^\mathsf{T}. \tag{4}$$

Here, $\mathbf{x}(i) \sim \mathcal{N}(\mu^*, \sigma^{*2}; \mathbb{R}_{>0})$ and $\mathbf{z}(i) \sim \mathcal{N}(\mu, \sigma^2; \mathbb{R}_{>0})$. Also, $\mu^*$ and $\sigma^{*2}$ are the parameters of the true distribution, and $\mu$ and $\sigma^2$ are functions of $\mathbf{v}$. In (4), $\mathbb{E}_\mathbf{z}[\mathbf{g}_\mathbf{z}(i)]$ admits a closed form expression (Johnson et al. (1995)), $\tilde{\mathbf{g}}_\mathbf{z} = \mathbb{E}_\mathbf{z}[\mathbf{g}_\mathbf{z}(i)] = \begin{bmatrix} -(\breve{\sigma}^2 + \breve{\mu}^2)/2 & \breve{\mu} \end{bmatrix}^\mathsf{T}$. Here, we define

$$\breve{\mu} = \mathbb{E}[\mathbf{z}(i)|0 < \mathbf{z}(i) < \infty] = \mu + \frac{\phi(-\mu/\sigma)}{1 - \Phi(-\mu/\sigma)}\sigma, \tag{5}$$

$$\breve{\sigma}^2 = \text{Var}(\mathbf{z}(i)|0 < \mathbf{z}(i) < \infty) = \sigma^2\left(1 - \frac{\mu}{\sigma}\frac{\phi(-\mu/\sigma)}{1 - \Phi(-\mu/\sigma)} - \left(\frac{\phi(-\mu/\sigma)}{1 - \Phi(-\mu/\sigma)}\right)^2\right), \tag{6}$$

where $\phi(\cdot)$ and $\Phi(\cdot)$ denote the probability density function and the cumulative distribution function of the standard normal distribution, respectively. We also need $\mathbb{E}_\mathbf{x}[\mathbf{g}_\mathbf{x}(i)]$ to compute $\mathbb{E}_{\mathbf{x},\mathbf{z}}[\mathbf{g}(i)]$. Since the value of $\mathbb{E}_\mathbf{x}[\mathbf{g}_\mathbf{x}(i)]$ is not known, we use the sample mean to estimate $\mathbb{E}_\mathbf{x}[\mathbf{g}_\mathbf{x}(i)]$. For this, all the positive samples are partitioned into $n_b$ batches. Let $S_j$ be the $j$-th batch. We then compute the vector $\overline{\mathbf{x}_j(i)} \in \mathbb{R}^2$ as follows:

$$\overline{\mathbf{x}_j(i)} = \frac{1}{|S_j|}\begin{bmatrix} \sum_{\mathbf{x}(i) \in S_j} \mathbf{x}(i)^2/2 & -\sum_{\mathbf{x}(i) \in S_j} \mathbf{x}(i) \end{bmatrix}^\mathsf{T}. \tag{7}$$

We then combine the $n_b$ vectors $\{\overline{\mathbf{x}_j(i)}\}_{j=1}^{n_b}$ to estimate $\mathbb{E}_{\mathbf{x},\mathbf{z}}[\mathbf{g}(i)]$. Recall that the observed samples are from the mixture distribution $\mathcal{D}_p(\mathbf{W}, \mathbf{b})$. Therefore, to compute the expectation with respect to the true distribution $\mathcal{D}(\mathbf{W}, \mathbf{b})$, a filter is applied on this set of $n_b$ vectors to mitigate the effect of the arbitrary outliers which is a common tool in the robust statistics literature and obtain $\mathbb{E}_\mathbf{x}[\mathbf{g}_\mathbf{x}(i)]$. We consider two filters: median and trimmed mean.

*Median:* The median based filter possesses the following robustness property. Typically, a median is the value separating the higher half from the lower half of a given set of points. If more than half of a given set of points are in $[-M, M]$ for some $M > 0$, then their median must be in $[-M, M]$. Thus, the median of $\{\overline{\mathbf{x}_j(i)}\}_{j=1}^{n_b}$ is the vector computed either from all the true samples or outliers whose magnitudes are comparable to the true samples.

*Trimmed Mean:* The trimmed mean removes the vectors among the set of $n_b$ vectors with relatively large and small values and computes the estimate of $\mathbb{E}_\mathbf{x}[\mathbf{g}_\mathbf{x}(i)]$ as the mean of the remaining vectors.

Here, we use the parameter $\beta$ to indicate the number of vectors to be discarded. This technique prunes the vectors $\overline{\mathbf{x}_j(i)}$, $j \in [n_b]$ that are computed from the batches with outliers having relatively high or low magnitudes. Therefore, we obtain the gradient estimate as

$$\tilde{\mathbf{g}}(i) = \tilde{\mathbf{g}}_{\mathbf{x}}(i) + \tilde{\mathbf{g}}_{\mathbf{z}}(i) = \text{filter}\left(\left\{\overline{\mathbf{x}_j(i)}\right\}_{j=1}^{n_b}\right) + \left[(\breve{\sigma}^2 + \breve{\mu}^2)/2 \quad \breve{\mu}\right]^{\mathsf{T}}, \tag{8}$$

where *filter* is either median or trimmed mean, and $\breve{\mu}$ and $\breve{\sigma}^2$ are given in (5) and (6), respectively.

Having computed the gradient estimate, we next present the proposed GD algorithm. In the $t$-th iteration, the algorithm performs the following three steps: The first step is *Gradient Computation Step* where the gradient $\tilde{\mathbf{g}}_t \in \mathbb{R}^2$ is computed from the observed positive samples using (8). The second step is *Update Step* where the gradient computed in the previous step is used to perform the GD update:

$$\tilde{\mathbf{v}}_t = \mathbf{v}_{t-1} - \gamma_{t-1}\tilde{\mathbf{g}}_{t-1}, \tag{9}$$

where $\gamma_{t-1} > 0$ is the diminishing step size. The last step is *Projection Step* where the objective function $\bar{l}(\mathbf{v})$ is a strongly convex function of $\mathbf{v}$, if $\mathbf{v}$ belongs to a bounded region. To control the strong-convexity of the objective function, we *project* the update vector $\mathbf{v}$ into the domain $D_r = \left\{\mathbf{v} \in \mathbb{R}^2 : 1/r \leq \mathbf{v}(1) \leq r, |\mathbf{v}(2)| \leq r\right\}$. Thus, the projection is

$$P(\mathbf{v}) = [\min\{\max\{\mathbf{v}(1), 1/r\}, r\} \quad \min\{\max\{\mathbf{v}(2), -r\}, r\}]. \tag{10}$$

The GD algorithm is run for $T$ iterations where $T$ is the algorithm parameter. The overall algorithm is summarized in Algorithm 2. This completes the first step of our algorithm that uses the GD algorithm to obtain the estimate of the bias vector and the row norms of the weight matrix.

Finally, using the estimates $\hat{\mathbf{b}}$ and $\hat{\boldsymbol{\Sigma}}$ obtained using the GD algorithm, we estimate $\hat{\theta}_{ij}$ similar to Wu et al. (2019) using (Williamson & Shmoys, 2011, Lemma 6.7). Specifically, we have

$$\hat{\theta}_{ij} = \pi - 2\pi\left(\frac{1}{n}\sum_{m=1}^{n}\mathbb{1}(\mathbf{x}_m(i) > \hat{\mathbf{b}}(i))\mathbb{1}(\mathbf{x}_m(j) > \hat{\mathbf{b}}(j))\right), \tag{11}$$

where $\mathbb{1}(\cdot)$ denotes the indicator function and $\hat{\mathbf{b}}$ is the estimate of the bias vector computed in the previous step using the projected GD algorithm.

The overall algorithm is given in Algorithm 1 where $\hat{\boldsymbol{\Sigma}}$ is the estimate of $\mathbf{W}\mathbf{W}^{\mathsf{T}}$. Here, Steps 1-5 estimate the row norms of $\mathbf{W}$ and $\mathbf{b}$, and Steps 6-9 estimate the angles between any two rows of $\mathbf{W}$.

Our algorithm is similar to that in Wu et al. (2019) except for the filter which is also a standard technique in robust statistics. However, our algorithm is different from the algorithm in Wu et al. (2019) in the following aspects. We do not rescale the unknown parameters (see Step 1 of Algorithm 2 in Wu et al. (2019)). We employ median or trimmed mean based filter to compute the full gradient and utilize the GD algorithm instead of SGD used in Wu et al. (2019). We note that our main contribution is the analysis rather than the algorithm which is presented in the next section.

## 4 THEORETICAL GUARANTEES FOR PARAMETER ESTIMATION

In this section, we provide our main result which characterizes the sample complexity for robust estimation of NN parameters. We address the general problem of learning the distribution generated by a NN in the presence of arbitrary outliers ($p \leq 1$). Hence, the proof techniques which have been devised for only the special case of $p = 1$ in Wu et al. (2019), are not applicable to our setup due to the presence of corrupted samples. Therefore, we develop novel proof techniques for performing the parameter estimation in the presence of arbitrary outliers. To arrive at the main result, we first present the error bounds for the two steps of Algorithm 1, which are described in Sec. 3.

The following proposition provides the estimation error bounds for the first step, i.e., the estimation of the bias vector $\mathbf{b}$ and the norms of the row vectors of $\mathbf{W}\mathbf{W}^{\mathsf{T}}$. The proof relies on the fact that the objective function is $\eta$-strongly-convex and $L$-smooth, and we show that the parameters $\eta, L$ are functions of $r$. Using the properties of the objective function, we the bound on the error between the true gradient $\nabla\bar{l}(\mathbf{v})$ and the estimated gradient $\tilde{\mathbf{g}}_t$ computed from the output samples with arbitrary outliers, obtained in terms of parameter $\epsilon$. This bound leads to the following result that shows that

we can estimate $\mathbf{b}(i)$ within an additive error of $\Xi\|\mathbf{W}(i,:)\|$ and $\mathbf{\Sigma}(i,i)$ within an additive error of $\Xi\|\mathbf{W}(i,:)\|^2$ where $\Xi$ is a function of the above parameters.

**Proposition 4.1.** *Suppose we initialize the projected gradient step (Algorithm 2) such that $\|\mathbf{v}_0 - \mathbf{v}^*\| = V$ and choose the step size $\gamma_t = \gamma$ and the number of iterations as $T$. Then, there exists $\gamma > 0$ such that for any $\delta, \epsilon \in (0,1)$, the output $(\hat{\mathbf{\Sigma}}(i,i), \hat{\mathbf{b}}(i))$ of the projected GD step of our algorithm with median-based filter for any $i \in [d]$ satisfies*

$$|\hat{\mathbf{\Sigma}}(i,i) - \|\mathbf{W}(i,:)\|^2| \leq \Xi\|\mathbf{W}(i,:)\|^2 \text{ and } |\hat{\mathbf{b}}(i) - \mathbf{b}(i)| \leq \Xi\|\mathbf{W}(i,:)\|, \tag{12}$$

*with probability at least $1 - \delta$ if the number of data samples $n \geq \frac{1}{p^2(p\Psi_\epsilon - 1/2)^2} \log \frac{1}{\delta}$. Here, the error bound is $\Xi = V\left(1 - \frac{L}{\eta+L}\right)^T + \frac{2\epsilon(L+\eta)}{L\eta}$ with $\eta, L > 0$ being constants that depend only on the parameter $r$. Also, $p$ is the probability of a sample being uncorrupted and $\Psi_\epsilon \in [1/2p, 1]$ is such that $O(\epsilon^3) \leq \Psi_\epsilon \leq O(\sqrt{\epsilon})$ with $\epsilon > O\left(p^{-1/3}\right)$ where the order constants depend only on $r$.*

The two terms in the error bound $\Xi$ in Proposition 4.1 can be interpreted as the following. The first term $V\left(1 - \frac{L}{\eta+L}\right)^T$ captures the convergence error due to the GD algorithm. For any upper bound on the convergence error $\Omega > 0$, Algorithm 1 runs in time $(\log \Omega - \log V)/\log\left(\frac{\eta}{\eta+L}\right)$. This relation shows that smaller convergence errors require more iterations. The second term $\frac{2\epsilon(L+\eta)}{L\eta}$ captures the error due to the gradient computed using the observed samples. Two factors contribute to this error: 1) difference between the (true) sample mean and the true mean, and 2) the arbitrary outliers. Also, the difference between the sample mean and true mean decreases with the number of samples. We notice that the sample complexity depends on $\Psi_\epsilon$ which is the probability of a sample being uncorrupted deviating from its mean and variance by $\epsilon$ and as $\Psi_\epsilon \leq O(\sqrt{\epsilon})$. Consequently, for smaller error $\epsilon$, we need a large number of samples $n = \Omega(\frac{1}{\epsilon p^4} \log \frac{d}{\delta})$. We also note that the lower bound on $\epsilon$ decreases with $p$.

The following proposition bounds the estimation error in the second step, i.e., the estimation of the angle between any two row vectors $\mathbf{W}(i,:)$ and $\mathbf{W}(j,:)$ (where $i \neq j \in [d]$). The result shows that we can estimate $\theta_{ij}$ within an additive error of $\Xi$.

**Proposition 4.2.** *Assume that (12) holds. Then, for a fixed pair of $i \neq j \in [d]$ and $\delta, \epsilon \in (0,1)$, the estimate $\hat{\theta}_{ij}$ of Algorithm 1 that uses a median-based filter satisfies*

$$|\cos\hat{\theta}_{ij} - \cos\theta_{ij}| < \sqrt{2}\pi p(p\Psi_\epsilon - 1/2) + 3\pi(1-p) + 2\pi(2-p)\Xi, \tag{13}$$

*with probability at least $1 - \delta$ if the number of data samples $n \geq \frac{1}{p^2(p\Psi_\epsilon - 1/2)^2} \log \frac{1}{\delta}$. Here, the error bound is $\Xi = V\left(1 - \frac{L}{\eta+L}\right)^T + \frac{2\epsilon(L+\eta)}{L\eta}$ with $\eta, L > 0$ being constants that depend only on the parameter $r$. Also, $p$ is the probability of a sample being uncorrupted and $\Psi_\epsilon \in [1/2p, 1]$ is such that $O(\epsilon^3) \leq \Psi_\epsilon \leq O(\sqrt{\epsilon})$ with $\epsilon > O\left(p^{-1/3}\right)$ where the order constants depend only on $r$.*

Combining Propositions 4.1 and 4.2, we next present the sample complexity needed to achieve a small parameter estimation error. Specifically, the diagonal entries, $\mathbf{\Sigma}(i,i)$, for $i \in [d]$ are obtained from Proposition 4.1. Further, the off-diagonal entries of $\mathbf{\Sigma}$ are obtained by combining the Propositions 4.1 and 4.2 to obtain the final result as follows:

**Theorem 4.3.** *Suppose we initialize the projected gradient step (Algorithm 2 with median-based filter) such that $\|\mathbf{v}_0 - \mathbf{v}^*\| = V$ and choose the step size $\gamma_t = \gamma$ and the number of iterations as $T$. Then, there exists $\gamma > 0$ such that for $\delta, \epsilon \in (0,1)$, the output of Algorithm 1, $(\hat{\mathbf{\Sigma}}, \hat{\mathbf{b}})$ satisfies*

$$\|\hat{\mathbf{\Sigma}} - \mathbf{W}\mathbf{W}^\mathsf{T}\|_F \leq \left(\sqrt{2}\pi p(p\Psi_\epsilon - 1/2) + \pi(2-p)(3+2\Xi)\right)\|\mathbf{W}\|_F^2 \tag{14}$$

$$\|\hat{\mathbf{b}} - \mathbf{b}\| \leq \Xi\|\mathbf{W}\|_F, \tag{15}$$

*with probability at least $1 - \delta$ if the number of data samples $n = \Omega(\frac{1}{\epsilon p^4} \log \frac{d}{\delta})$. Here, $p$ is the probability of a sample being uncorrupted and $\Psi_\epsilon \in [1/2p, 1]$ is such that $O(\epsilon^3) \leq \Psi_\epsilon \leq O(\sqrt{\epsilon})$ with $\epsilon > O\left(p^{-1/3}\right)$ where the order constants depend only on $r$.*

**Table 1:** Runtime of various schemes when $p = 0.95$, $n = 20000$, and $d = 5$. The table indicates that SGD schemes are faster than their GD counterparts.

| Scheme | Oracle | Without Filter | With Median | With Trimmed Mean |
|--------|--------|----------------|-------------|-------------------|
| **GD** | 16.95 s | 17.73 s | 34.44 s | 60.78 s |
| **SGD** | 1.24 s | 1.60 s | 2.11 s | 3.62 s |

We note that in Propositions 4.1 and 4.2, and Theorem 4.3, $\eta$, $L$ and the other order constants depend on the parameter $r > 0$. Recall that the parameter $r$ is required for the convexity region $D_r$ and we can choose $r$ to be a large constant. Thus, $r$ is a relatively insensitive parameter which does not require any hand-tuning.

As the probability of a sample being uncorrupted $p$ reduces, we observe that the error bound in (14) becomes larger as expected. Also, as the probability of a sample being uncorrupted decreases, the number of samples $n$ required for the error bound increases. Moreover, the obtained error bounds increase with the problem dimension for a fixed number of samples.

Our results are also comparable with the existing error bounds for ReLU parameter estimation without outliers from (Wu et al., 2019, Theorem 1). Specifically, if there are no outliers, for any $\zeta, \delta \in (0, 1)$, the number of samples $n = O(\frac{1}{\zeta^2} \log \frac{d}{\delta})$ are sufficient for the SGD algorithm in Wu et al. (2019) to achieve the following error bounds with probability at least $1 - 2\delta$,

$$\|\hat{\mathbf{\Sigma}} - \mathbf{W}\mathbf{W}^\mathsf{T}\|_F \leq \zeta\|\mathbf{W}\|_F^2 \text{ and } \|\hat{\mathbf{b}} - \mathbf{b}\| \leq \zeta\|\mathbf{W}\|_F, \tag{16}$$

where $\zeta$ indicates the parameter estimation error. Comparing the above bounds with (14) and (15), we observe that our bounds are larger due to the presence of arbitrary outliers. In particular, $\Xi$ consists of two error terms. The first term $V\left(1 - \frac{L}{\eta + L}\right)^T < 1$ in our error bounds results from the convergence of the GD algorithm and is similar to $\zeta < 1$ in (16). However, the second error term $\frac{2\epsilon(L+\eta)}{L\eta}$ quantifies the error due to the gradient estimated from the output samples with arbitrary outliers and is not present in (16). Additionally, in (14), the extra term $3\pi(2 - p)$ further increases the bound compared to the error bound in (16). Therefore, the error $\zeta$ in (16) can not be directly compared to the error in (14) and (15). Further, our sample complexity depends on the probability of a sample being uncorrupted $p$ and the second error term $\epsilon$ whereas the complexity in (Wu et al., 2019, Theorem 1) depends on the square of the convergence error term $\zeta$ in their sample complexity. Also, when there is no outlier ($p = 1$), the two sample complexities do not match.

## 5 SIMULATION RESULTS

In this section, we provide numerical results to verify the performance of our algorithm. In our setup, the columns of $\mathbf{W}$ are chosen as the left singular vectors of random matrices from the standard Gaussian distribution. For $\mathbf{b}$, we use a random vector from the standard normal distribution whose negative values are replaced with zeros. The mixture of samples are generated such that a sample comes from $\mathcal{D}(\mathbf{W}, \mathbf{b})$ with probability $p$ and from $\mathcal{G} = \mathcal{N}(5, 1)$ with probability $1 - p$. The hyperparameters in our proposed algorithm are set as $r = 3$ and $\gamma_t = \frac{1}{0.1t}$. We use the batch-splitting approach to compute the gradient. This approach induces randomization. As the output samples consist of a fraction of arbitrary outliers, the standard gradient descent algorithm without any filter would perform poorly as shown in our results. Also, we choose the total number of GD and SGD iterations $T$ to be $|\mathcal{X}_{>0}|/100$ where $|\mathcal{X}_{>0}|$ is the number of positive output samples. From our experiments, we observe that the errors first decrease with the number of iterations and then after a certain number of iterations, the errors flatten. Based on this observation, we chose the number of iterations (see Appendix A.4). Note that we discard the zero entries and only consider only the set of positive samples $\mathcal{X}_{>0}$ for the number of iterations as they do not convey any information about the row norms of $\mathbf{W}$ and bias vector $\mathbf{b}$.

We compute two error metrics from the estimated parameters and the ground truth, $\|\hat{\mathbf{\Sigma}} - \mathbf{W}\mathbf{W}^\mathsf{T}\|_F/\|\mathbf{W}\|_F^2$ and $\|\hat{\mathbf{b}} - \mathbf{b}\|_2/\|\mathbf{W}\|_F$. Also, we compare our algorithm with two other schemes: oracle scheme (estimation using the true samples only), and scheme without filter.

The results are shown in Figs. 1 and 2, and Table 1, and the observations from them are as follows.

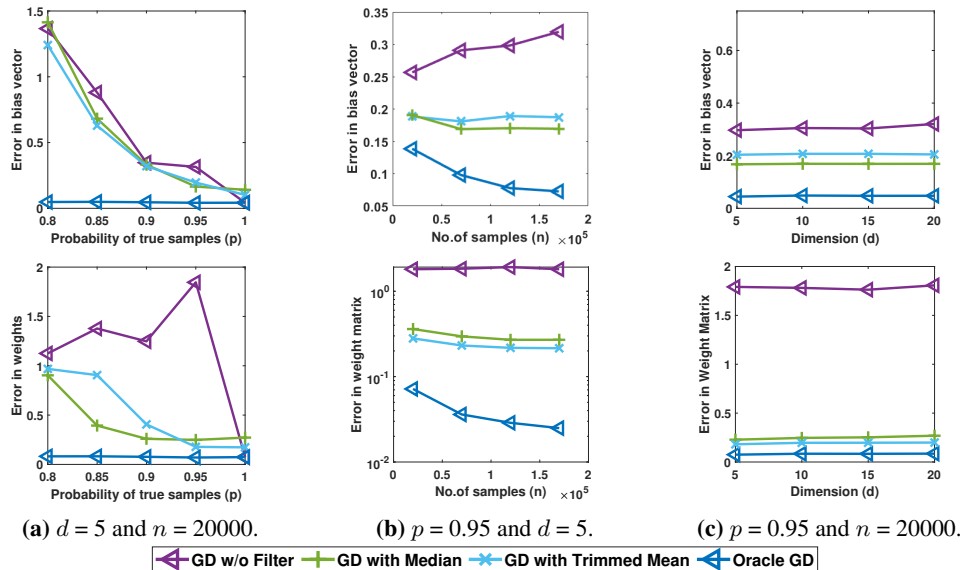

**Figure 1:** Comparison of the different GD schemes as a function of $p$ (first column), $n$ (second column), and $d$ (third column). The figures indicate that utilizing the filters improves the performance of GD algorithm for mixture of samples.

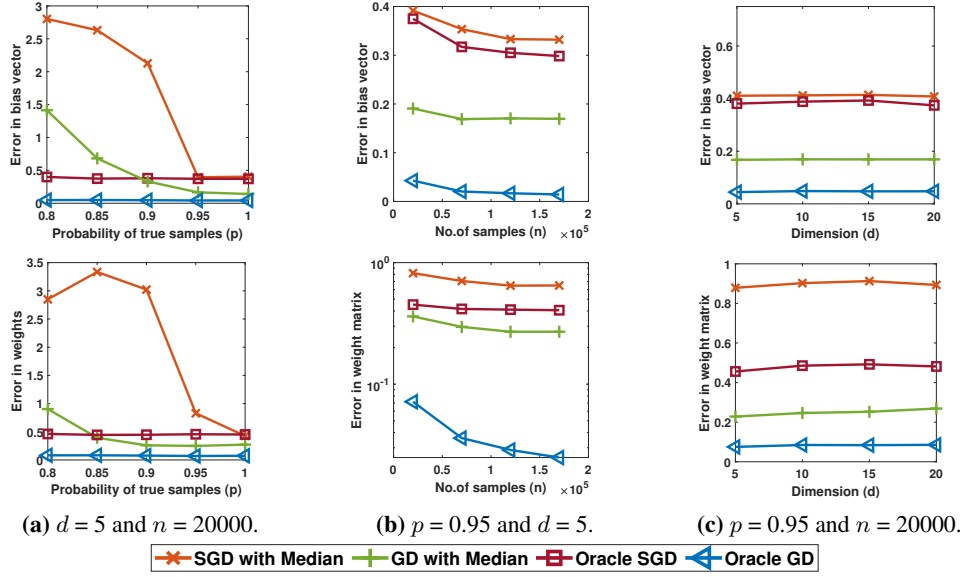

**Figure 2:** Comparison of GD and SGD schemes as a function of $p$ (first column), $n$ (second column), and $d$ (third column). We infer that SGD is more sensitive to corrupted samples compared to GD.

*Effect of filters:* From Fig. 2, we observe that the GD schemes perform better than the corresponding SGD schemes. Further, the performance is improved when we use median based filter along with GD or SGD (for instance, as seen by comparing the performances of GD without Filter and GD with Median in Fig. 1). Thus, the filter ensures that the effect of the outliers is reduced, and the curves are closer to the Oracle GD scheme. Finally, from Fig. 1, we also infer that the median-based approach performs slightly better than the trimmed mean-based approach.

*Dependence on probability of a sample being uncorrupted $p$:* The variation in the error estimation as a function of $p$ is shown in Figs. 1a and 2a. The performance of the proposed schemes improve with $p$, as expected. This trend is because as $p$ increases, the fraction of outliers in the observed samples

decreases, which leads to better performance. Note that the oracle schemes assume the knowledge of true samples. Thus, their performance does not change with $p$. On the contrary, the performance of the schemes without filters and SGD schemes with filter do not have a monotonic behavior with $p$. This observation shows that these schemes are not able to handle the outliers effectively. Further, all the schemes converge to the corresponding oracle schemes when $p = 1$.

*Dependence on number of samples $n$:* Figs. 1b and 2b show how the estimation error of different schemes varies with $n$. We observe that the parameter estimation errors computed using Oracle SGD and Oracle GD schemes decrease as the number of samples increases. This trend is because the oracle schemes consider only the true samples. Also, the Oracle GD scheme performs better than the Oracle SGD scheme. Further, the estimation errors computed using our proposed schemes with median and trimmed mean based filters perform better than GD without filter. Hence, the filters mitigate the effect of arbitrary outliers and having more samples reduces the estimation errors.

*Dependence on dimension $d$:* In Figs. 1c and 2c, we plot the error in different schemes as a function of the dimension $d$. We note that due to the presence of arbitrary outliers, the errors increase as the dimension increases for GD without filter. Further, we observe a performance improvement using our proposed schemes with median and trimmed mean-based filters compared to the other benchmarking schemes, supporting our previous conclusions that the filters mitigate the effect of outliers. However, we observe a slow increase in the parameter estimation errors using our proposed schemes with median and trimmed mean based filters as we increase the dimension for fixed number of samples. We observe that the errors when using the oracle schemes also increase slowly as the dimension is increased (Wu et al. (2019)).

*Comparison of runtimes:* From Table 1, we observe that the SGD schemes run faster than their GD counterparts. This is because the SGD utilizes only one sample for the gradient whereas GD utilizes all the samples for the gradient. Further, the usage of median and trimmed mean based filters along with SGD or GD increases their computation times. However, we reiterate that using the GD scheme with filters offers the best performance in terms of error (see Fig. 2). Further, the runtimes of the trimmed mean-based schemes are significantly higher than those of the median-based schemes.

To summarize, from our results, we conclude that GD schemes offer better performance than the SGD schemes. Also, we deduce that compared to the trimmed mean-based scheme, the median-based filter is computationally more efficient while offering similar or lower estimation error. Moreover, the median-based scheme is parameter free (the trimmed mean depends on the parameter $\beta$) and enjoys strong theoretical guarantees (see Theorem 4.3). Overall, the median-based GD algorithm is the most effective approach to estimate the NN parameters in the presence of the outliers and performs slightly better than trimmed mean-based GD algorithm. Further, we observe that GD schemes with filters perform better than Oracle SGD scheme.

## 6 CONCLUSION

In this paper, we proposed an algorithm for estimation of the parameters of a single-layer ReLU neural network from the truncated Gaussian samples where each sample was assumed to be an arbitrary outlier with a fixed probability. Our only assumption was that the bias vector was non-negative. We derived a GD-based algorithm for parameter estimation and incorporated filters to handle the outliers. We analyzed the sample complexity of the proposed algorithm in terms of the parameter estimation error. The efficacy of our approach was also demonstrated using numerical experiments. Removing the non-negativity assumption on the bias vector is a promising direction for future work. Also, extending our results to multi-layer neural networks is also an interesting problem to consider in the future.

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
