# OpenReview forum: "LEARNING DISTRIBUTIONS GENERATED  BY SINGLE-LAYER RELU  NETWORKS  IN  THE PRESENCE  OF ARBITRARY OUTLIERS"
_ICLR.cc/2022/Conference — ICLR 2022 Submitted_

### Official Review · Reviewer_D2X8 · 2021-10-29

**Correctness:** 3
**Technical Novelty And Significance:** 2
**Empirical Novelty And Significance:** Not applicable
**Recommendation:** 3
**Confidence:** 3

**Main Review:**

This is a well-defined problem and the authors make an improvement on previous works on learning the parameters in generative models, which all (to my knowledge) avoided the existence of outliers.  At a technical level, the argument has nearly-exact parallels with the procedure of Wu et al., with a small modification for accoutning for outliers that involves swapping the gradient estimate the median and trimmed mean.  It appears to me that the current result does not recover the Wu et al. result as the probability p of uncorruption tends to 1, which is weird (see below; please correct me if I'm wrong).  The presentation also had some issues making it difficult to determine how accurate the parameter estimation really is.  These are the main reasons I'm leaning towards rejection.

A few concrete comments I'm hoping the authors can address:

(1) Is it the case that the current result does not recover the parameters as p --> 1?  In Theorem 4.3, for p=1, the second term doesn't appear to go to zero, since one will be left with a 3 pi ||W||_F^2 term.  Compare this with Theorem 1 of Wu et al.

(2) The statements of Theorem 4.3 and Propositions 4.1/4.2 seem incomplete.  Where is the quantifier for epsilon?  The appearance of the \Psi terms is also confusing and not really remarked upon in any detail, although they seem key, and the lack of placement of epsilon in the statement of the results makes it harder to tell what is going on.   This may be related to the fact that the discussion following the statement of Theorem 4.3 talks about the "first term" and the "second term", but the quantities mentioned do not actually appear as direct quantities in the theorem statement.

(3) The appendix is structured oddly.  There are no formal statements of lemmas/theorems; there are only "Steps" presented on page 12, and then claimed proofs for the "steps" in the subsequent sections.  This made it hard to verify the veracity of many of the claims.  There should be precise, formal statements of results so that the reader can verify the proofs.

For instance: in Appendix A.1.3. it seems that the authors are only proving the result for the median filter (assuming this is what med{.} denotes), although the results are claimed for both the trimmed mean and the median filter.  Is there a proof for the trimmed mean somewhere?  Can the authors give some more details on how precisely (47)-(50) are derived?

(4) The authors should be more explicit about how the estimation procedure nearly identically mimics that of Wu et al., with the difference being the outlier-robust mean estimation.  If I had not previously read the Wu et al. paper, it would not have been clear from the text how indebted the present work is to Wu et al.


Some other minor comments/typos encountered while reading:

* p.1, "...datasets impacted by a Byzantine" -- not sure what this refers to?

* p.2, "adversial"

* p.3, "RVe x"

* Regarding the assumption that the bias is negative: Claim 2 of Wu et al. shows that having negative bias would require exponential sample complexity.

* Where is \tilde g_x defined in algorithm 2?

*p. 9 and experiments: why is it that the p=0.95 case is so bad for standard gradient descent?  This seems unusual.  Presumably this is using the batch-splitting approach, so there is randomization induced by which samples appear first in the algorithm.  Did the authors average over different realizations of the batches?

* In the appendix, g_t appears everywhere but seems to refer to \tilde g_t.  Are these the same?




**Summary Of The Paper:**

The authors study the problem of learning the parameters of a generative model defined by a single-layer relu network in the presence of outliers from the Huber contamination model.  They derive a gradient-based algorithm for estimating the norms and angles between rows of the generating matrix, which suffices for estimating the matrix WW^T (which is the only estimable parameters in this setting).  Their analysis is largely the same as that of Wu et al. '19, with additional care needed to take care of outliers.  This is done via trimmed mean or median estimates.


**Summary Of The Review:**

This is a well-defined problem and the authors make an improvement on previous works on learning the parameters in generative models, which all (to my knowledge) avoided the existence of outliers.  At a technical level, the argument has nearly-exact parallels with the procedure of Wu et al., with a small modification for accoutning for outliers that involves swapping the gradient estimate the median and trimmed mean.  It appears to me that the current result does not recover the Wu et al. result as the probability p of uncorruption tends to 1, which is weird (see below; please correct me if I'm wrong).  The presentation also had some issues making it difficult to determine how accurate the parameter estimation really is.  These are the main reasons I'm leaning towards rejection.

---

> ### Author Response · Authors · 2021-11-22
> **We thank the reviewer for their comments. Due to space constraint, we have given a shorter versions of the questions and brief answers to them. A more detailed discussion addressing the comments is provided in the updated paper.**
>
> Q1: The current result does not recover the Thm.1 of Wu et al. as $p \rightarrow 1$? Compare the two results.
>
> A1: The second term does not go to zero because the proof techniques which have been devised for the special case of $p =1$ in Wu et al. (2019), are not applicable to our setup where $p\leq 1$. So, our paper develops novel proof techniques. Consequently, the error term $\zeta$ from (16) can not be directly compared to the error in (14) and (15). See the updated discussion in the last paragraph of Sec. 4.
>
> Q2: Where is the quantifier for epsilon and $\Psi$? The quantities mentioned in the discussion following Thm. 4.3 do not actually appear as direct quantities in the theorem statement.
>
> A2: In our results, $\epsilon$ is not the actual estimation error but it represents the error due to the gradient estimated from the output samples with arbitrary outliers. $\epsilon$ captures the tradeoff between estimation error and sample complexity which also depends upon $\Psi_{\epsilon}$ (function of $\epsilon$). In Prop. 4.1, the estimation error bounds depend on $\Xi$ which captures the convergence error (depends on $T$) due to the GD algorithm and the error due to the gradient computed using the observed samples
> (depends on $\epsilon$). In Prop. 4.2, the angle estimation error bound is in terms of both $\Psi_{\epsilon}$ and $\Xi$, both are functions of $\epsilon$. Further, combining the Props. 4.1 and 4.2, we obtain the final result as shown in Thm. 4.3 which states that the overall error bounds depend on both $\Psi_{\epsilon}$ and $\Xi$ that are functions of $\epsilon$ (see the updated last paragraphs of Sec. 4).
>
> Q3: The appendix is structured oddly with no formal statements of lemmas/theorems. In App. A.1.3. it seems that the authors are only proving the result for the median filter, although the results are claimed for both the trimmed mean and the median filter. Is there a proof for the trimmed mean somewhere? Can the authors give some more details on how precisely (47)-(50) are derived?
>
> A3: We note that the technique of providing the proof as various steps is a standard practice in the literature (Joseph et al. 2021: Proof of Thm. 3.1 in App. B). Our results provided in Props. 4.1 and 4.2, and Theorem 4.3 are derived for median based filter denoted by med(.) in (45) (explicitly mentioned in the revised theorem statements). Regarding the Eqs. (45)-(48), they directly follow from the definition of $g_t$ and the standard formula for the first and second moments of the truncated Gaussian distribution. Please refer App. A.1.3 of the updated paper for a detailed discussion.
>
> Q4: The authors should be more explicit about how the estimation procedure nearly identically mimics that of Wu et al., with the difference being the outlier-robust mean estimation.
>
> A4: We agree with the reviewer that our algorithm is similar to that in Wu et al. (2019) except for the filter which is also a standard technique in robust statistics. We note that our main contribution is the sample complexity required for robust estimation. To address the reviewer's comments, we have shortened the discussion on the algorithm development and explicitly mentioned the ideas from Wu et al. (2019). We have only retained the discussion on the steps of our algorithm that are different from the existing algorithm and removed the steps that are common in both the algorithms in Sec. 3.
>
> Q5: p.1, what  does `datasets impacted by a Byzantine' refer to?
>
> A5: We consider the model where the set of output samples consists of a fraction of arbitrary outliers. This model can be interpreted as dataset of samples being corrupted by an adversary (Byzantine) where each sample can be corrupted with a fixed probability.
>
> Q6: p.2, `adversial', and p.3, `RVe x'
>
> A6: Corrected.
>
> Q7: Regarding the assumption that the bias is negative: Claim 2 of Wu et al. shows that having negative bias would require exponential sample complexity.
>
> A7: Wu et al. (2019) showed that if $\mathbf{b}\in \mathbb{R}^d$, exponentially large number of samples are required to estimate the bias. This holds true in our case as in the presence of arbitrary outliers if the bias has large negative values, we would require larger number of samples to estimate the bias compared to that in Wu et al. (2019). So, we assume that the bias is non-negative.
>
> Q8: Where is \tilde g_x defined in Alg. 2?
>
> A8: Defined in Step 2 of Alg. 2 and (8) of Sec. 3.1.
>
> Q9: p. 9: why is it that the p=0.95 case is bad for standard gradient descent? Presumably this is using the batch-splitting approach, so there is randomization induced by which samples appear first in the algorithm. Did the authors average over different realizations of the batches?
>
> A9: We use batch-splitting approach. As the output samples consist of a fraction of arbitrary outliers, the standard gradient descent algorithm without any filter would perform poorly as shown in our simulation results.
>
> Q10:  Are g_t and \tilde g_t the same?
>
> A10: Yes.

---

> > ### Author Response · Authors · 2021-11-22
> > **We include the references cited earlier in our response.**
> >
> > Reference:
> > 1. Wu et al., Learning distributions generated by one-layer ReLU networks. Advances in Neural Information Processing Systems, 32:8107–8117, 2019.
> > 2. Joseph et al.,  Control-lability of network opinion in Erdos–Renyi graphs using sparse control inputs. SIAM Journal on Control and Optimization, 59(3):2321–2345, 2021.

---

> > ### Comment · Reviewer_D2X8 · 2021-11-30
> > **thanks**
> >
> > ** Updated review post author-response **
> >
> > I've read the authors' response as well as the other reviewers and their responses.  Unfortunately, I'm going to decrease my score to a 3-reject.  I don't find the authors' response to reviewer concerns about not recovering the noiseless recovery result in the p --> 1 limit convincing.  Given the paper's limited technical contributions in the proof -- namely, utilize the Wu et al. argument together with standard robust estimation procedures to deal with Huber contamination -- in order for me to recommend acceptance, it is necessary for the result to at least recover the Wu et al. result in the p --> 1 limit.

---

### Official Review · Reviewer_CkRT · 2021-10-31

**Correctness:** 4
**Technical Novelty And Significance:** 2
**Empirical Novelty And Significance:** 1
**Recommendation:** 3
**Confidence:** 3

**Main Review:**


* In the bound in equation (16), the multiplier is at least $3\pi$. The error does not tend to 0 even as the sample size $n\rightarrow 0$. This is true even there are no outliers $(p=1)$.
* In Figure 1(b), the errors for GD with filters do not decrease with increasing sample size. Why is that?
* For how many iterations was SGD run? The number of GD iterations is set to $|\mathcal{X}_{>0}|/100$. Is this a sufficient number of iterations? Can you please show/tell how the error goes down with iterations, say for $n=20000, d=5, p=0.9$? Can you please explain why you discard examples with zero entries to set the number of iterations?


Minor:
* The terms $\eta, L$ used in Proposition 4.2 should be described further in the main text. Otherwise, it is not possible to understand the results. Why is there a matrix transpose in the definition of $\Xi$?
* The similarity between the bounds in equation (18) and results in Theorem 4.3 is superficial. This is because, the bounds in (18) can be tightened arbitrarily close to 0 by increasing the sample size, but the bounds in Theorem 4.3 cannot be.






**Summary Of The Paper:**

Given observations $x_1, \cdots, x_n$ assume that a fraction $p$ of the observations are drawn from
$$ x = ReLU(Wz + b), \text{ where } z \sim N(0, I_k) $$
and the rest are drawn from an arbitrary distribution $\mathcal{G}.$
The authors study the problem of estimating $W, b$ in this model.  They assume that $b$ has non-negative entries.
The authors separate the problem in two parts:
* estimate $\| W(i,:) \|, b_i$ from the $i$th coordinates of the observations
* estimate the angles between the rows of $W$
They use a gradient descent algorithm for the first part and use a clever random projections based idea for the second part.
 They derive error bounds of their method and demonstrate the performance of their method through simulations.





**Summary Of The Review:**

The error bounds do not seem tight and there are some concerning issues with their simulations.

---

> ### Author Response · Authors · 2021-11-22
> **We thank the reviewer for their comments. The reviewer was mainly concerned with the convergence of our bounds and suggested some simulations with respect to number of iterations. We address the reviewer's concerns as below and have also made changes to the main draft.**
>
> Q1: In the bound in (16), the multiplier is at least $3\pi$. The error does not tend to 0 even as the sample size $n\rightarrow 0$. This is true even there are no outliers $(p=1)$.
>
> A1: Assuming that the reviewer meant $n\rightarrow \infty$, we agree with the reviewer that the error in Thm. 4.3 (in (14) and (15)) does not go to zero as we increase the number of samples. In our work, we address a general problem of learning the distribution generated by a NN in the presence of arbitrary outliers ($p\leq 1$). Hence, the proof techniques which have been devised for the special case of $p = 1$ in Wu et al. (2019), are not applicable to our setup due to the presence of corrupted samples. Therefore, we developed novel proof techniques for performing the parameter estimation in the presence of arbitrary outliers. Our main contribution is the sample complexity for robust estimation. Using our proof techniques, the error bounds thus obtained in the presence of arbitrary outliers ($p<1$) do not go to zero.
>
> Q2: In Figure 1(b), the errors for GD with filters do not decrease with increasing sample size. Why is that?
>
> Q2: We thank the reviewer for this comment. In general, having more samples provides more information. However, increasing the number of samples also gives more corrupted samples which may cancel out the potential gain achieved by the increase in the number of samples. However, we plot the estimation error curves by tuning the step size parameter for different values. We observed that by applying a smaller value of the step size, the errors for GD with filters decrease with increasing number of samples.
>
> Q3: For how many iterations was SGD run? The number of GD iterations is set to $|\mathcal{X}_{>0}|/100$. Is this a sufficient number of iterations? Can you please show/tell how the error goes down with iterations, say for $n=20000,d=5,p=0.5$? Can you please explain why you discard examples with zero entries to set the number of iterations?
>
> A3: The SGD algorithm was also run for $|X_{>0}|/100$ iterations. Regarding the sufficiency of iterations, from our experiments, we observe that the errors first decrease with the number of iterations and then after a certain number of iterations, the errors flatten. Based on this observation, we chose the number of iterations. About the discarding of zero entries, they do not convey any information about the row norms of $\mathbf{W}$ and bias vector $\mathbf{b}$. Thus, the input to the learning algorithm (SGD or GD) which estimates the row norms and bias vector is the set of positive samples $X_{>0}$.
>
> Q4: The terms $\eta,L$ used in Prop. 4.2 should be described further in the main text. Otherwise, it is not possible to understand the results. Why is there a matrix transpose in the definition of $\Xi$?
>
> A4: The term $\eta$ is the strong convexity parameter and $L$ is the smoothness parameter. We would like to clarify that $T$ in the first term of the parameter $\Xi$ denotes the number of iterations and not matrix transpose.
>
> Q5: The similarity between the bounds in (18) and results in Thm. 4.3 is superficial. This is because, the bounds in (18) can be tightened arbitrarily close to 0 by increasing the sample size, but the bounds in Thm. 4.3 cannot be.
>
> A5: We agree that the bounds in (16) can be tightened arbitrarily close to zero by increasing the sample size whereas ours cannot be. However, in our work, we address a general problem of learning the distribution generated by a NN in the presence of arbitrary outliers ($p\leq 1$). Hence, the proof techniques which have been devised for the special case of $p =1$ in Wu et al. (2019), are not applicable to our setup due to the presence of corrupted samples. Therefore, we developed novel proof techniques for performing the parameter estimation in the presence of arbitrary outliers. Further, we note that the error $\zeta$ in (16) (from \cite{wu2019learning}) can not be directly compared to the error in (14) and (15) (in Thm. 4.3) due to following reason. The $\zeta$ in (16) (from \cite{wu2019learning}) indicates the parameter estimation error. However, $\Xi$ in our work consists of two error terms. The first term $V\left( 1-\frac{ L }{\eta+ L }\right)^{T}$ results from the convergence of the GD algorithm. The second term is a function of $\epsilon$ that quantifies the error due to the gradient estimated from the output samples with arbitrary outliers.
>
> Note that the eq. nos. are according to the updated draft.
>
> Reference:
> 1. Wu et al., Learning distributions generated by one-layer ReLU networks. Advances in Neural Information Processing Systems, 32:8107–8117, 2019.

---

> > ### Comment · Reviewer_CkRT · 2021-11-28
> > **Thanks to the authors for addressing my comments.**
> >
> > The authors addressed most of my questions. However the results are worse than those in Wu et al. for the special case p=1. So, the overall results are not tight and need further work, in my opinion.

---

### Official Review · Reviewer_ApGT · 2021-11-02

**Correctness:** 3
**Technical Novelty And Significance:** 3
**Empirical Novelty And Significance:** Not applicable
**Recommendation:** 5
**Confidence:** 3

**Details Of Ethics Concerns:**

N/A.

**Main Review:**

Strengths
+ The paper is well written.
+ The techniques in this paper are fairly natural and intuitive (for the specific assumptions made in this paper -- see below.)
+ The results make sense and are consistent with existing results for learning single-layer NNs without outliers.

Weaknesses
- The algorithm (rather strongly) depends on the i.i.d. Gaussianity assumption of the input data, which -- of course -- is unrealistic in practice. I admit that it is not uncommon to see such assumptions in theory-focused papers, but that's a larger criticism beyond this paper. For this particular case however, it's hard to envision how the techniques would generalize to more realistic situations -- the Williamson-Shmoys angle estimates according to (11) only hold true for i.i.d. Gaussian inputs.
- I'm a bit confused about the claim about learnability. The main results are for learning WW' where W is the weight matrix, but in the intro the authors talk about sample complexity of learning the network parameters. How could that translate to learning W up to epsilon relative error (and how would sample complexity be affected in this case?)
- I'm a bit confused about the utility of Algorithm 2 (and Proposition 4.1). Isn't it just univariate Gaussian estimation with outliers, and if yes, why propose a new method and not just use existing robust estimators?
- It may be helpful to clarify the 'r' hyperparameter in Algorithm 2, and how it affects the constants in Prop 4.1 and the sample complexity.

---
Thanks for the response. Keeping score unchanged.

**Summary Of The Paper:**

The paper presents and analyzes an approach to learn single-layer ReLU networks (assuming Gaussian-distributed input data) in the Huber contamination model where a constant fraction of the training samples may be corrupted.

The approach involves two steps:
* estimating row-wise weight norms using gradient descent with median/trimmed mean gradients.
* estimating angles between rows using standard properties of inner products with Gaussian vectors.



**Summary Of The Review:**

This is a well-written theoretical paper on understanding the sample complexity of learning shallow ReLU neural nets from corrupted samples; however, both the approach and the obtained results are somewhat narrowly applicable to the case of Gaussian inputs and do not seem like they can generalize to more challenging settings.

---

> ### Author Response · Authors · 2021-11-22
> **We thank the reviewer for their comments. The reviewer was mainly concerned with the Gaussianity assumption and the learnability of the parameters. We address the reviewer's concerns as below and have also made changes to the main draft.**
>
> Q1: The algorithm depends on the iid Gaussianity assumption of the input data, which is unrealistic in practice yet not an uncommon assumption in theory-focused papers. In this case, it's hard to envision how the techniques (eg: angle estimation) would generalize to more realistic situations.
>
> A1: We agree with the reviewer that the proposed algorithm depends on the assumption that the input is Gaussian. We consider the Gaussianity assumption for mathematical tractability as mentioned by the reviewer. Further, this assumption helps us in considering the Williamson-Shmoys angle formula for obtaining the angle estimate error bounds. The Gaussianity assumption is motivated by a popular generative approach to estimate a high-dimensional distribution from observed samples in the case of image generation by the generative adversarial networks Goodfellow et al. (2014); Radford et al. (2015); Arjovsky et al. (2017).
>
> Q2: How would the main results on learning WW', translate to learning W and how would sample complexity be affected in this case?
>
> A2: We recall that our goal is to learn a neural network that can model the underlying distribution of a given set of samples. Learning the distribution is equivalent to learning $\mathbf{W}$ and $\mathbf{b}$ such that $(\mathbf{W}\mathbf{W}^{\mathsf{T}}$  is the covariance matrix and $\mathbf{b})$ is the mean of the underlying Gaussian distribution. But, the weight matrix $\mathbf{W}$ may not be identifiable from the distribution $\mathcal{D}(\mathbf{W},\mathbf{b})$. Specifically, $(\mathbf{W},\mathbf{b})$ and $(\mathbf{W}\mathbf{Q},\mathbf{b})$ define the same distribution for any unitary matrix $\mathbf{Q}$. Since our goal is to learn the distribution, learning either $\mathbf{W}$ or $\mathbf{W}\mathbf{Q}$ is sufficient. The identifiability problem of $\mathbf{W}$ does not affect the estimation of distribution. In short, we focus on the learnability of the underlying distribution and not the learnability of the neural network parameters.
>
> Q3: I'm a bit confused about the utility of Alg. 2 (and Prop. 4.1). Isn't it just univariate Gaussian estimation with outliers, and if yes, why propose a new method and not just use existing robust estimators?
>
> A3: We agree with the reviewer that we employ a two step procedure along with the trimmed mean or median filter to remove the outliers which is a common tool in the robust statistics literature. Our main contribution is the sample complexity for robust estimation rather than the algorithm itself. Prop. 4.1 provides the estimation of the bias vector $\mathbf{b}$ and the norms of the row vectors of $\mathbf{W}\mathbf{W}^{\mathsf{T}}$. In Prop. 4.1, the estimation error bounds have been obtained in terms of $\Xi$ which consists of two terms as follows. The first term $\left( 1-\frac{ L }{\eta+ L }\right)^{T}V$ captures the convergence error due to the GD algorithm. The second term is a function of $\epsilon$ that captures the error due to the gradient computed using the observed samples. The two factors that contribute to this error are 1) difference between the sample mean and the true mean, and 2) the arbitrary outliers. We also obtain the sample complexity which is a function of $\Psi_{\epsilon}$. Recall that $n\geq \frac{1}{p^2(p\Psi_{\epsilon}-1/2)^2}\log\frac{1}{\delta}$ and as $\Psi_{\epsilon} = O(\sqrt{\epsilon})$, we get the sample complexity as $n=\Omega(\frac{1}{{\epsilon}p^4}\log\frac{d}{\delta})$. We have only retained the discussion on the steps of our algorithm that are different from those of the algorithm in Wu et al. (2019) and removed the discussion about the steps which are common in Sec. 3.
>
> Q4: It may be helpful to clarify the `r' hyperparameter in Alg. 2, and how it affects the constants in Prop 4.1 and the sample complexity.
>
> A4: The role of the hyperparameter $r$ in Alg. 2 is the following. Strong convexity of  the objective function  is a desired property used in the analysis of the sample complexity, and one way to ensure strong convexity is to project the update vector $\mathbf{v}$ onto a bounded region. We denote this region as $D_r$ that is parameterized by $r > 0$, $D_r = (\mathbf{v}\in\mathbb{R}^2: 1/r\le \mathbf{v}(1) \le r, |\mathbf{v}(2)|\leq r)$. So, $r$ controls the strong-convexity parameter of the objective function. Specifically, from App. A.1.1 (from (26)), a large value of $r$ would lead to a small strong-convexity parameter $\eta$ as $\eta = \Omega(\min( 1/r,1/r^2 ))$. Also, the smoothness parameter $L$ is $L=O(r^8)$, from App. A.1.1 (from (33)). Regarding how $r$ affect the results, we note from Prop. 4.1 that the number of samples is given by $n\geq \frac{1}{p^2(p\Psi_{\epsilon}-1/2)^2}\log\frac{1}{\delta}$. Further, from App. A.1.4, the upper bound for parameter $\Psi_{\epsilon}$ in terms of $r$ is obtained as $O(r^2)$ (from (58)-(62)). Therefore, the number of samples, $n= \Omega(1/r^4)$.
>
> Note that the eq. nos. are according to the updated draft.

---

> > ### Author Response · Authors · 2021-11-22
> > **We include the references cited earlier in our response.**
> >
> > References:
> > 1. Goodfellow et al., Generative adversarial networks. arXiv preprint arXiv:1406.2661, 2014.
> > 2. Radford et al., Unsupervised representation learning with deep convolutional generative adversarial networks. arXiv preprint arXiv:1511.06434, 2015.
> > 3. Arjovsky et al., Wasserstein generative adversarial networks. ´In International Conference on Machine Learning, pp. 214–223. PMLR, 2017.
> > 4. Wu et al., Learning distributions generated by one-layer ReLU networks. Advances in Neural Information Processing Systems, 32:8107–8117, 2019.
> > 5. Daskalakis et al., Efficient statistics, in high dimensions, from truncated samples. In 2018 IEEE 59th Annual Symposium on Foundations of Computer Science (FOCS), pp. 639–649. IEEE, 2018.

---

> > ### Comment · Reviewer_ApGT · 2021-11-30
> > **Thanks for the replies**
> >
> > and in particular answering Q4.
> >
> > Re Q1 - I unfortunately don't agree with the analogy to gaussian noise inputs for GAN models.
> >
> > Keeping my score unchanged.

---

### Official Review · Reviewer_fTQC · 2021-11-03

**Correctness:** 4
**Technical Novelty And Significance:** 2
**Empirical Novelty And Significance:** Not applicable
**Recommendation:** 3
**Confidence:** 4

**Main Review:**

- The problem of learning ReLU Gaussian distribution over outliers that this work is looking at is not very well motivated and I am not convinced by the significance of the problem.
- The techniques used in this paper look incremental in addition to [1] where they worked with the same problem but without outliers. This paper essentially follows the same two step procedure as in [1] and use a robust gradient estimation procedure over it which is also a common tool used in the robust statistics literature.
- The main theorem - Theorem 4.3 statement is really hard to understand. Particularly, I see that the error does not go down to 0 even with infinite number of samples and time. There is a pi(2-p)3 term which does not go down to 0. Also, [1] proved a lower bound of 1/epsilon^2 but this paper gives an upper bound of 1/epsilon? Can the authors please clarify this part? In general, I think the writing of the paper could be improved.

**Summary Of The Paper:**

The paper studies the problem of parameter estimation of one layer ReLU neural networks where the data is generated by the same network and with probability p, some of the examples are corrupted to arbitrary outliers.They give an algorithm with theoretical guarantees on the sample complexity and also, provide simulations on how the estimation error scales with the important parameters.

**Summary Of The Review:**

I would make a recommendation for rejecting this paper.

---

> ### Author Response · Authors · 2021-11-22
> **We thank the reviewer for their comments. The reviewer was mainly concerned with the significance of the problem of learning the distribution generated by a NN in the presence of arbitrary outliers and the techniques used. We address the reviewer's concerns as below and have also made changes to the main draft.**
>
> Q1.1: The problem of learning ReLU Gaussian distribution over outliers that this work is looking at is not very well motivated and I am not convinced by the significance of the problem.
>
> A1.1: The problem of learning the distribution of a given dataset is motivated by the wide use of neural network-based generative models, and learning with corrupted data is a well-studied problem in the supervised learning setting (Bakshi et al. (2019), Goel et sl. (2019), Mukherjee et al. (2020), Zhang et al. (2019), Frei et al. (2020), Vempala et al. (2019)). Here, the corruption of a sample could be due to an adversary which can influence the NN to generate false outputs. Also, the Gaussianity assumption is motivated by a popular generative approach to estimate a high-dimensional distribution from observed samples in the case of image generation by generative adversarial networks (Goodfellow et al. (2014), Radford et al. (2015), Arjovsky et al. (2017)). Our problem of estimating the parameters of the distribution generated by a NN from noisy samples was mentioned as an open problem in (Wu et al. (2019)). Hence, we address the open problem of learning from a corrupted dataset by providing a robust algorithm that can estimate the row norms of the weight matrix and bias vector and analyzing the performance of the proposed algorithm that estimates the weight matrix and the bias vector.
>
> Q1.2: The techniques used in this paper look incremental in addition to [1] where they worked with the same problem but without outliers. This paper essentially follows the same two step procedure as in [1] and use a robust gradient estimation procedure over it which is also a common tool used in the robust statistics literature.
>
> A1.2: We agree with the reviewer that we employ a two step procedure similar to that in (Wu et al. (2019)) along with the trimmed mean or median filter to remove the outliers which is a common tool in the robust statistics literature. Our main contribution is the sample complexity for robust estimation rather than the algorithm itself. We establish that the algorithm in (Wu et al. (2019)) can be made robust to outliers if we employ a median based or trimmed mean based filter. Further, we bound the additional number of samples required for robust estimation. Note that the technique for obtaining the error bounds for row norms of weight matrix and bias vector in (Wu et al. (2019)) directly follows from (Daskalakis et al. (2018)). However, the presence of corrupted samples makes the analysis of parameter estimation non-trivial, and does not directly follow from the existing results. In particular, we derived a novel bound for the error between the true gradient and the estimated gradient from the output samples, which is a crucial step in obtaining the estimation error bounds in Proposition 4.1. Moreover, we provide empirical evidence to validate the theoretical results. We have only retained the discussion on the steps of our algorithm that are different from those of the algorithm in (Wu et al. (2019)) in Section 3.
>
> Q1.3: The main theorem - Theorem 4.3 statement is really hard to understand. Particularly, I see that the error does not go down to 0 even with infinite number of samples and time. There is a $\pi(2-p)3$ term which does not go down to 0. Also, [1] proved a lower bound of $1/\epsilon^2$ but this paper gives an upper bound of $1/\epsilon$? Can the authors please clarify this part? In general, I think the writing of the paper could be improved.
>
> A1.3: We agree with the reviewer that the error in Theorem 4.3 (in (14) and (15)) does not go to zero as we increase the number of samples or the number of iterations. In our work, we address a general problem of learning the distribution generated by a NN in the presence of arbitrary outliers ($p\leq 1$). Hence, the proof techniques used in (Wu et al. (2019)) are not applicable to our setup due to the presence of corrupted samples as they have been derived for the special case of $p=1$. Therefore, we developed novel proof techniques for estimating the parameter (weight matrix and bias vector) in the presence of arbitrary outliers. Our main contribution is the sample complexity for robust estimation. Using our proof techniques, the error bounds thus obtained are loose (applicable for the general case of $p\leq 1$) compared to those obtained in (Wu et al. (2019)) (applicable only to the special case of $p=1$).
>
> Regarding the lower bound in (Wu et al. (2019)), a direct comparison of the sample complexity results is not valid due to the following. The $\epsilon$ in the sample complexity result of (Wu et al. (2019)) indicates the parameter estimation error. However, in our work, $\epsilon$ is not the actual estimation error but it quantifies an error in the intermediate step of the algorithm. Specifically, it represents the error due to the gradient estimated from the output samples with arbitrary outliers.
>
> Note that the eq. nos. are according to the updated draft

---

> > ### Author Response · Authors · 2021-11-22
> > **We include the references cited earlier in our response.**
> >
> > References:
> > 1. Bakshi et al., Learning two layer rectified neural networks in polynomial time. In Conference on Learning Theory, pp. 195–268. PMLR, 2019.
> > 2. Goel et al., Time/accuracy tradeoffs for learning a ReLU with respect to gaussian marginals. arXiv preprint arXiv:1911.01462, 2019.
> > 3. Mukherjee et al., Guarantees on learning depth-2 neural networks under a data-poisoning attack. arXiv preprint arXiv:2005.01699, 2020.
> > 4. Zhang et al., Learning one-hidden-layer ReLU networks via gradient descent. In The 22nd International Conference on Artificial Intelligence and Statistics, pp. 1524–1534. PMLR, 2019.
> > 5. Frei et al., Agnostic learning of a single neuron with gradient descent. arXiv preprint arXiv:2005.14426, 2020.
> > 6. Vempala et al., Gradient descent for one-hidden-layer neural networks: Polynomial convergence and SQ lower bounds. In Conference on Learning Theory, pp. 3115–3117. PMLR, 2019.
> > 7. Goodfellow et al., Generative adversarial networks. arXiv preprint arXiv:1406.2661, 2014.
> > 8. Radford et al., Unsupervised representation learning with deep convolutional generative adversarial networks. arXiv preprint arXiv:1511.06434, 2015.
> > 9. Arjovsky et al., Wasserstein generative adversarial networks. ´In International Conference on Machine Learning, pp. 214–223. PMLR, 2017.
> > 10. Wu et al., Learning distributions generated by one-layer ReLU networks. Advances in Neural Information Processing Systems, 32:8107–8117, 2019.
> > 11. Daskalakis et al., Efficient statistics, in high dimensions, from truncated samples. In 2018 IEEE 59th Annual Symposium on Foundations of Computer Science (FOCS), pp. 639–649. IEEE, 2018.

---

### Decision · Program_Chairs · 2022-01-20

**Decision:**

Reject

**Comment:**

This paper studies the problem of learning single-layer neural networks under Gaussian marginals in the presence of outliers. The authors give a recovery algorithm in this setting. The consensus among the reviewers was that the paper lacks technical depth. Specifically, the algorithm is a minor tweak of the one in Wu et al. 2019 for the case without outliers. Another concern was that the algorithm does not recover the prior result when the fraction of uncorrupted points goes to 0. Overall, the paper is below the acceptance threshold.